# PosS: Position Specialist Generates Better Draft for Speculative Decoding

## Abstract

Speculative decoding accelerates Large Language Model (LLM) inference by using a small draft model to predict multiple tokens, and a large target model to verify these tokens in parallel. Recent studies leverage the hidden state of the target model to enhance draft model prediction accuracy. However, existing methods suffer from the degrading quality of draft token predictions at later positions, due to error accumulation in draft model generated features. In this paper, we propose Position Specialists (**PosS**), which consist of multiple position-specialized draft layers to generate tokens at assigned position(s). Position specialists substantially improve token acceptance rate at later positions within each drafting round, as each specialist only needs to focus on handling a certain level of draft model feature deviation. Experiment results on Llama-3-8B-Instruct and Llama-2-13B-chat across six datasets demonstrate that **PosS** effectively improves over baselines on average acceptance length and speed-up ratio. Our codebase is available at `https://github.com/poss-speculative-decoding/Position-Specialist`.

## 1 Introduction

Speculative decoding (Leviathan et al., 2022; Chen et al., 2023) is an effective approach to accelerate the autoregressive decoding of Large Language Models (LLMs) through a draft-then-verify framework. Specifically, it employs a lightweight draft model to generate candidate tokens autoregressively, which are then verified by the larger target model in parallel to determine accepted tokens from proposed draft tokens, thereby reducing overall decoding time. The effectiveness of speculative decoding largely depends on the average acceptance length $\tau$ (accepted token counts per round) from the prediction depth $L$ (predicted token counts generated by the draft model per round).

Recent efforts (Cai et al., 2024; Li et al., 2024a;b; 2025b) in speculative decoding utilize the target model hidden states as input to enhance draft model prediction accuracy. EAGLE (Li et al., 2024a;b; 2025b) employs a one-layer Transformer as the draft model and trains it to predict the next token with features from the target model. However, EAGLE-1,2 exhibit a training–inference discrepancy: target model features are always available during training, but sometimes not at inference time. Instead, it relies on features generated by the draft model. HASS (Zhang et al., 2024) and EAGLE-3 (Li et al., 2025b) partially address this discrepancy by training the draft model to predict the next token with features from previous draft steps. However, both approaches suffer from relying on a single draft model to predict tokens at multiple positions in the draft sequence.

We hypothesize that **effective draft model should be position-specialized** within the prediction length $L$: early positions require accurate predictions with reliable target model features, while later positions must learn to mitigate the increasing levels of feature deviations. To evaluate the prediction quality across positions, we introduce the metric of position-wise acceptance rate (pos-acc) to measure the conditional probability of accepting the $i^{\text{th}}$ token given the acceptance of its preceding $(i-1)^{\text{th}}$ token. Our analysis reveals that both EAGLE and HASS suffer from rapidly degrading pos-acc beyond the first few predicted tokens. This confirms our hypothesis that a single draft model is limited by its generalization capability of various positions.

To address this challenge, we propose Position Specialists (**PosS**), a novel framework that consists of multiple position-specialized draft layers, called position specialists. Each position specialist is trained for predicting tokens at its assigned position(s), and only needs to handle an expected level

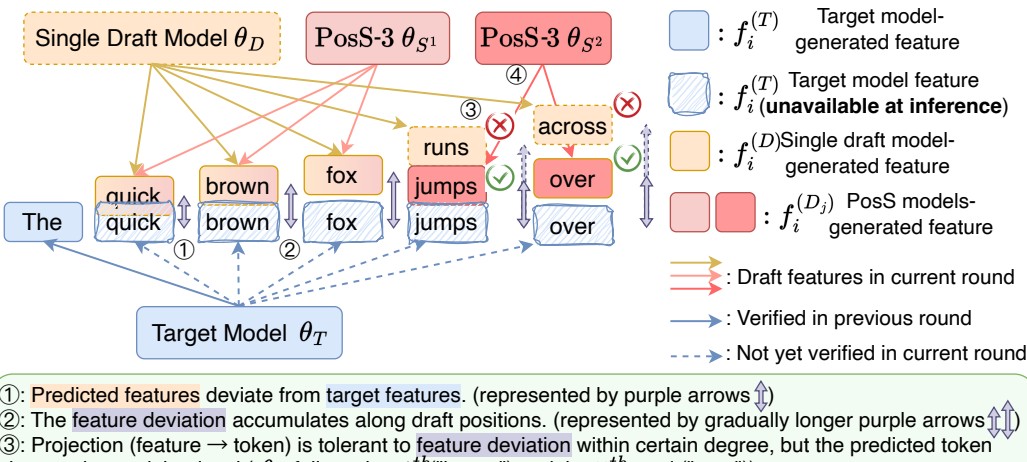

Figure 1: This figure illustrates how **POSS** improves over single draft models like EAGLE and HASS at inference time. Draft model-generated features have two functions: generate draft tokens via LM head projection and serve as input to draft model at the next position. When generating draft tokens, a slight deviation between draft and target features does not change the predicted tokens. However, when biased features are used to predict the next positions, the existing deviation is amplified and causes a larger deviation in following steps, eventually leading to wrong token prediction. **POSS**, however, resets the deviation propagation to a low level by switching to another draft layer. As a result, **POSS** maintains draft token accuracy at later positions, achieving better acceptance rate.

of feature deviation at that position, thus enabling more accurate draft token predictions than a single draft model which needs to handle varying levels of feature deviation at different positions.

We conduct extensive experiments on two model sizes (Llama-3-8B-Instruct and Llama-2-13B-chat) across six benchmark datasets, and demonstrate that **POSS** consistently outperforms baseline methods. On the average of 6 test datasets, **POSS** surpasses the strong baseline EAGLE-3 on average acceptance length by 9.2% (from 4.69 to 5.12) and on speed-up ratio by up to 10.5% (from 2.96x to 3.27x). We also carry out a comprehensive analysis and reveal that **the efficiency of POSS comes from reduced rounds of speculative generation**, as a higher position-wise acceptance rate at deeper positions enables longer acceptance length $\tau$ per round.

Our primary contributions include:

- We introduce position-wise acceptance rate (pos-acc) as a crucial metric for analyzing the draft quality of speculative decoding approaches.

- We propose Position Specialists (**POSS**), a novel framework that employs position-specialized layers to address the challenge of accumulated levels of feature deviation in draft predictions.

- We conduct extensive experiments and analysis to demonstrate that **POSS** outperforms baseline methods on both average acceptance length and speed-up ratio.

## 2 PRELIMINARY

### 2.1 SPECULATIVE DECODING

Speculative decoding harnesses the principle of speculative execution (Kung & Robinson, 1979), where a smaller, faster draft model $\theta_D$ works alongside a larger target language model $\theta_T$ that we aim to accelerate. The standard speculative decoding (Leviathan et al., 2022) operates in three key phases. First, the draft model $\theta_D$ autoregressively generates a candidate sequence of length $L$. Next, the target model $\theta_T$ evaluates all $L$ draft tokens in parallel with a single forward pass. Finally,

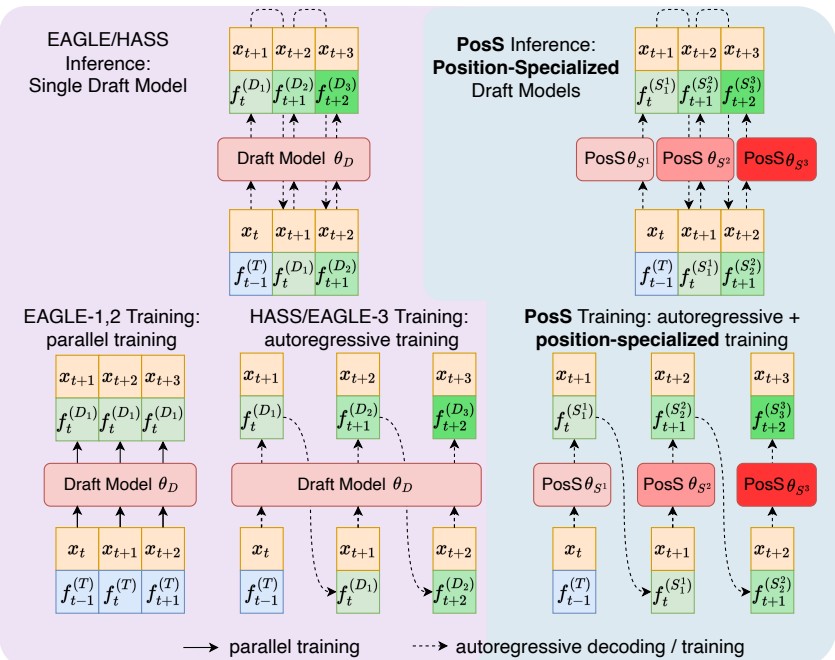

Figure 2: The inference and training stages of EAGLE, HASS, and our **POSS** method. The dashed lines represent autoregressive decoding or training, and the solid lines represent parallel training. The input concatenates context word embeddings $x$ and features from previous step $f$. During **inference**, EAGLE and HASS use a single draft model $\theta_D$ to generate features $f^{(D_i)}$ for each position $i$ recursively. For **draft model training**, EAGLE-1,2 uses the target model feature $f^{(T)}$ as input for training. HASS and EAGLE-3 additionally use draft model-predicted features $f^{(D_i)}$. Different from them, **POSS** introduces different position specialists $\theta_{S^j}$. During inference, the position-specialized draft models autoregressively generate features $f^{(S_i^j)}$, where position $i$ corresponds to the specialist $\theta_{S^j}$. At training stage, **POSS** applies position-specialized training: A specialist $\theta_{S^j}$ is trained on the $i^{\text{th}}$ position using the previous step specialist feature.

draft tokens that align with the target distribution are accepted. This parallel evaluation significantly reduces inference latency compared to traditional token-by-token generation.

## 2.2 HIDDEN STATE ASSISTED SPECULATIVE DECODING

Recent research efforts (Cai et al., 2024; Li et al., 2024a;b; 2025b) discover the potential of the target model's hidden state. Instead of using a complete auxiliary model for drafting, researchers demonstrate that applying a few extra layers to process the last-layer hidden states of the target model, referred to as features, suffices for effective draft generation. Medusa (Cai et al., 2024) uses multiple language model heads to project a feature vector into different output spaces to predict several subsequent tokens simultaneously. EAGLE-1,2 (Li et al., 2024a;b) represent a significant breakthrough in speculative decoding through concatenating input embedding with feature vectors. EAGLE-3 (Li et al., 2025b) substitutes the last-layer hidden states with those from low, middle, and high-level layers, further improving the performance. EAGLE family employs a one-layer Transformer as the draft model $\theta_D$ and reuses LM head of the target model for token prediction. At generation step $t$, EAGLE's draft model $\theta_D$ predicts the next token $x_{t+1}$ based on context $x_{\leq t}$ and features $f_{<t}$:

$$P(x_{t+1}) = \text{Head}(\theta_D([x_t; f_{t-1}^{(T)}], [x_{t-1}; f_{t-2}^{(T)}], \ldots, [x_1; f_0^{(T)}]))$$

(1)

Figure 2 provides an example of EAGLE at inference stage. $\theta_D$ autoregressively generates draft tokens $x_{t+1}, x_{t+2}, x_{t+3}$, where the subscripts represent the timesteps. Inputs are derived from dif-

ferent sources, denoted by superscripts: $f^{(T)}$ represents feature from the target model; $f^{(D_i)}$ represents feature from the $i^{th}$ draft step of the draft model $D$. $f^{(D)}$ is used instead of $f^{(T)}$ when the target model features are unavailable, before the forward pass completion of subsequent tokens. Therefore, the prediction of the $k^{th}$ draft position is formulated as:

$$P(x_{t+k}) = \text{Head}(\theta_D([x_{t+k-1}; f_{t+k-2}^{(D_{k-1})}], \dots, [x_{t+1}; f_t^{(D_1)}], [x_t; f_{t-1}^{(T)}], \dots, [x_1; f_0^{(T)}])) \quad (2)$$

Specifically, equation 2 degenerates to equation 1 when $k = 1$.

Although EAGLE-1,2 perform inference with equation 2, it is solely trained on equation 1. This exhibits a fundamental training-inference discrepancy: $\theta_D$ needs to predict the subsequent tokens ($k > 1$) with its own generated features during inference, but it never observes its own prediction errors during training, which impairs the ability to effectively predict long draft sequences.

HASS and EAGLE-3 explicitly address the discrepancy through recursive feature alignment in training. Therefore, the training process aligns with the inference process, as shown in Figure 2. Eventually, they improve the acceptance probabilities of tokens at later positions compared to EAGLE-2.

## 3 METHOD

In this section, we introduce our Position Specialist (**PoSS**) approach for speculative decoding. We first introduce the concept of position-wise acceptance rate to reveal the fundamental limitations in existing approaches in Section 3.1. We then propose our **PoSS** with position specialized training in Section 3.2 to address the limitation.

### 3.1 POSITION-WISE ACCEPTANCE RATE

Previous speculative decoding frameworks rely heavily on the generalizability of a single draft layer for multi-position token generation. EAGLE-1,2 trains $\theta_D$ only on the immediate next position but expects it to generalize to subsequent positions at inference time. While HASS and EAGLE-3 train $\theta_D$ on both the immediate and later positions, only one draft model is used to generalize across diverse feature sources and different draft positions. As the draft model is a single Transformer layer, the generalizability is inherently limited due to model capacity.

To demonstrate the generalization limitation of EAGLE and HASS, we introduce **position-wise acceptance rate** (**pos-acc**), which measures the probability that a token at position $i$ is accepted given its preceding token at position $i-1$ is accepted. The **pos-acc** at position $i$ is defined as:

$$\textbf{pos-acc}_i = P(A_i \mid A_{i-1}) = \frac{P(A_{i-1} \cap A_i)}{P(A_{i-1})} = \frac{P(A_i)}{P(A_{i-1})} , \quad i > 1 \quad (3)$$

where $A_i$ denotes the event that the token at position $i$ is accepted during the verifying process. Notice that the target model acceptance follows a strict sequential dependency: if $x_i$ is accepted, its preceding tokens $x_{[0:i-1]}$ must also have been accepted, and therefore $A_i \subset A_{i-1}$.

We point out that higher **pos-acc** is crucial for achieving a higher acceptance length $\tau$ at each draft-verification round. For a draft sequence of length $L$, the probability of accepting all draft tokens up to position $k$ ($k \leq L$) is:

$$P(A_k) = P(A_1 \cap A_2 \cap \dots \cap A_k) = \begin{cases} P(A_1) & \text{if } k = 1 \\ P(A_1) \prod_{i=2}^{k} \textbf{pos-acc}_i & \text{if } k > 1 \end{cases} \quad (4)$$

This chain rule decomposition reveals that the overall acceptance length depends on the multiplication of **pos-acc**, and is particularly sensitive to degradation in any single position. Notably, token prediction becomes increasingly challenging at later positions due to the accumulation of prediction errors and the growing uncertainty in longer draft positions.

In Figure 3, we demonstrate the empirical **pos-acc** of EAGLE-2,3 and HASS. EAGLE-2's **pos-acc** deteriorates rapidly beyond position $k = 1$. This is because the draft model of EAGLE-2 is solely

trained on predicting the next immediate token. HASS and EAGLE-3 are able to maintain relatively higher **pos-acc** at later positions because a single draft model is trained on multiple subsequent positions. However, their **pos-acc** at position $k = 1$ becomes lower than other methods by about $1\%$ to $2\%$, because of their compromise to other positions. This critically impairs the overall acceptance length due to the multiplicative nature of the acceptance probability in equation 4.

## 3.2 Position Specialists Improve Position-Wise Acceptance Rate

To address the aforementioned limitation, we introduce Position Specialists (**POSS**) to preserve early-position acceptance rate while enhancing later position predictions. **POSS** consists of multiple position-specialized draft layers, called position specialists. Each specialist is trained for certain position(s) and generates draft tokens at its assigned position(s). The number of positions that a specialist is assigned to can be pre-defined as $n$, and **POSS**-$n$ means each specialist is responsible for $n$ positions. Figure 2 exhibits the training and inference of **POSS**-1. In the example, there are 3 position specialists $\{\theta_{S^i}\}_{i=1}^3$, with each assigned to predict the draft token $x_{t+i}$. During training, each specialist $\theta_{S^i}$ learns to predict using the input feature of draft model at the previous step.

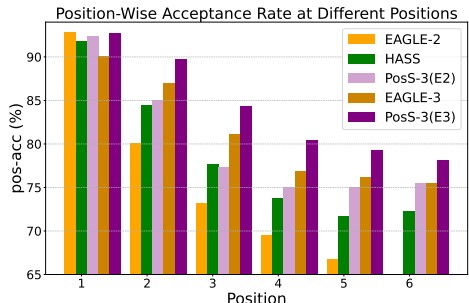

Figure 3: Position-wise acceptance rate (**pos-acc**) of the $i^{th}$ token on MT-Bench dataset by various speculative decoding methods. The **pos-acc** of EAGLE-2 and HASS decays fast as the draft sequence gets longer. Our proposed **POSS** method keeps a stable and higher **pos-acc** even at the deepest position .

Figure 1 illustrates the draft process at the inference stage, showing the deficit of a single draft model and how **POSS** improves over it. All methods with EAGLE frameworks require the features from the target model for drafting. During a draft phase, the target model-generated features at most positions are unavailable because these draft tokens have not been verified by the target model. In this case, the input of the draft model is substituted by draft model-generated features, which have an inevitable deviation from target features. Although the slightly deviated features may still predict the current token accurately, the bias is passed to the next draft position because these features also serve as input to the draft model for the next token prediction. The feature deviation propagates and accumulates along draft positions until it becomes too big to yield a correct token. Existing work, HASS and EAGLE-3, tries to mitigate feature deviation by aligning training and inference, as exhibited in Figure 2. However, the limited capacity of a single draft model prohibits it from handling all deviations. As a result, they either perform worse at later or earlier positions.

The key point of **POSS** is to improve draft models' ability to handle all kinds of deviations. Unlike the single draft model, each position specialist in **POSS** is position-aware. The draft layers beyond the first one are only trained to make accurate predictions from biased feature input, which enables them to mitigate the deviation accumulated from previous steps. Another benefit brought by position-specialized draft layers is that it avoids conflict optimization directions. The lower **pos-acc** of EAGLE-3 at the first position is likely a result of training on all positions. Since it trains a lot on draft model-generated features, with different levels of deviations, as input, it performs worse when the input is only target features. However, this is not a problem for **POSS**, because tasks of largely different optimization directions are distributed to different position specialists.

We further highlight that **POSS** is orthogonal to EAGLE-2 and EAGLE-3 frameworks. **POSS** uses the same loss as HASS on EAGLE-2 framework and the same loss as EAGLE-3 on EAGLE-3 framework, but optimizes our designed draft model architecture. **POSS**-n predicts draft token $x_{t+k}$ with equation 5, which differs from equation 2 only in the superscripts.

$$P(x_{t+k}) = \text{Head}(\theta_{(S^{\lceil k/n \rceil})}([x_{t+k-1}; f_{t+k-2}^{(S^{\lceil (k-1)/n \rceil})}], \ldots, [x_{t+1}; f_t^{(S^1)}], [x_t; f_{t-1}^{(T)}], \ldots, [x_1; f_0^{(T)}])) \tag{5}$$

For implementation, we conduct an experiment in Section 5.1, comparing **POSS**-1,2,3, where they show similar **pos-acc**. Considering the extra memory usage, the setting **POSS**-3 is recommended.

## 4 EXPERIMENT

Table 1: Average acceptance length $\tau$ of all methods. L3 8B represents Llama-3-8B-Instruct, L2 13B represents Llama-2-13B-Chat.

| Model | Method | MT-Bench | Alpaca | GSM8K | Natural Questions | CNN/DM | HumanEval | Avg. |
|---|---|---|---|---|---|---|---|---|
| | | | | | Temperature=0 | | | |
| | EAGLE-2 | 4.11 | 4.32 | 4.25 | 3.38 | 3.61 | 4.70 | 4.06 |
| | HASS | 4.42 | 4.62 | 4.61 | 3.54 | 3.92 | 5.20 | 4.39 |
| | Gumiho | 4.27 | 4.19 | 4.59 | 3.58 | 3.84 | 5.18 | 4.28 |
| L3 8B | PoSS-3(E2) | **4.52** | **4.82** | **4.81** | **3.64** | **4.05** | **5.41** | **4.54** |
| | EAGLE-3 | 4.73 | 5.07 | 4.89 | 3.71 | 4.18 | 5.55 | 4.69 |
| | EAGLE-3+HASS | 3.79 | 3.74 | 3.85 | 3.05 | 3.21 | 4.61 | 3.71 |
| | PoSS-3(E3) | **5.15** | **5.50** | **5.43** | **4.13** | **4.54** | **5.95** | **5.12** |
| | EAGLE-2 | 4.86 | 4.64 | 5.01 | 4.15 | 4.30 | 5.78 | 4.79 |
| L2 13B | HASS | 5.28 | 5.16 | 5.40 | 4.43 | 4.59 | 6.37 | 5.21 |
| | Gumiho | 4.78 | 4.55 | 4.97 | 4.13 | 4.41 | 5.82 | 4.78 |
| | PoSS-3(E2) | **5.33** | **5.17** | **5.48** | **4.52** | **4.70** | **6.43** | **5.27** |
| | | | | | Temperature=1 | | | |
| | EAGLE-2 | 3.83 | 4.15 | 4.09 | 3.18 | 3.39 | 4.50 | 3.86 |
| | HASS | 4.01 | 4.39 | 4.49 | **3.40** | 3.65 | 5.00 | 4.16 |
| | Gumiho | 3.90 | 3.95 | 4.33 | 3.32 | 3.59 | 4.84 | 3.99 |
| L3 8B | PoSS-3(E2) | **4.13** | **4.46** | **4.67** | 3.37 | **3.76** | **5.12** | **4.25** |
| | EAGLE-3 | 4.31 | 4.62 | 4.75 | 3.45 | 3.85 | 5.30 | 4.38 |
| | EAGLE-3+HASS | 3.21 | 3.34 | 3.53 | 2.44 | 2.89 | 4.29 | 3.28 |
| | PoSS-3(E3) | **4.66** | **4.98** | **5.24** | **3.81** | **4.14** | **5.69** | **4.75** |
| | EAGLE-2 | 4.69 | 4.44 | 4.82 | 4.12 | 4.25 | 5.54 | 4.64 |
| L2 13B | HASS | 5.04 | 4.92 | 5.24 | **4.36** | **4.60** | 6.03 | 5.03 |
| | Gumiho | 4.57 | 4.40 | 4.80 | 4.03 | 4.25 | 5.66 | 4.62 |
| | PoSS-3(E2) | **5.12** | **4.98** | **5.39** | 4.35 | 4.54 | **6.15** | **5.09** |

### 4.1 EXPERIMENT SETUP

**Metrics.** We evaluate the performance of our approach using two key metrics: speed-up ratio and average acceptance length.

- **Speed-up Ratio**: The speed-up ratio measures the improvement in generation efficiency compared to the vanilla target model decoding, calculated as the ratio between throughputs (tokens generated per second) of a speculative decoding approach to that of the target model autoregressive decoding. A higher speed-up ratio indicates better performance.

- **Average Acceptance Length** $\tau$: The average acceptance length represents the mean number of tokens accepted in each round of $L$ drafting positions (denoted as prediction length). It reflects how effectively the draft model can predict longer sequences that match the target model output. Longer acceptance lengths generally correlate with improved efficiency as they reduce the number of draft iterations needed.

**Datasets.** We conduct comprehensive experiments on six datasets, following EAGLE. This includes MT-Bench (Zheng et al., 2023) for multi-turn conversation, Alpaca (Taori et al., 2023) for instruction following, GSM8K (Cobbe et al., 2021) for mathematical reasoning, Natural Questions (Kwiatkowski et al., 2019) for question answering, CNN/Daily Mail (shortened to CNN/DM) (Nallapati et al., 2016) for summarization, and HumanEval (Chen et al., 2021) for code generation.

**Target Models.** We evaluate on two model sizes: Llama-3-8B-Instruct (L3 8B) and Llama-2-13B-chat (L2 13B). This allows us to evaluate how our approach performs across model sizes. Llama-3-8B-Instruct serves as our primary model for ablation studies and detailed analysis, while Llama-2-13B demonstrates the scalability of our method to larger models.

**Draft Methods.** We evaluate the following methods for comparison. **EAGLE-2**: the base method in EAGLE-2 framework, trained with a classification loss on token and a regression loss on feature. **HASS**: EAGLE-2 with recursive feature alignment training and a topk token distillation loss. **Gumiho** Li et al. (2025a): drafting the first two positions with EAGLE-2 and the following positions with Medusa. **PoSS(E2)**: our method with the loss of HASS. **EAGLE-3**: the base method in EAGLE-3 framework with recursive feature alignment training, trained only with the classification loss on token. **EAGLE-3+HASS**: EAGLE-3 with all training strategies of HASS. **PoSS(E3)**: our method with the loss of EAGLE-3.

**Implementations.** Our implementation is built upon the open-source repositories of EAGLE[1], HASS[2], and SpecForge[3]. As EAGLE-2 is a widely adopted method and HASS is built upon it, we mainly experiment with the EAGLE-2 framework. Besides, we also experiment **PoSS**-3 on the recently introduced EAGLE-3 framework in our Llama-3-8B-Instruct setting for fair comparison. To distinguish our method on two frameworks, they are named "**PoSS**(E2)" and "**PoSS**(E3)" when needed. Because EAGLE-3 introduces a much larger training set, we reproduce it using similar training steps as methods in EAGLE-2 framework for fair comparison. All models apply tree-draft inference implemented by EAGLE-2,3. The detailed settings are introduced in Appendix A.

Table 2: Speed-up ratios of all methods. L3 8B represents Llama-3-8B-Instruct, L2 13B represents Llama-2-13B-Chat.

| Model | Method | MT-Bench | Alpaca | GSM8K | Natural Questions | CNN/DM | HumanEval | Avg. |
|---|---|---|---|---|---|---|---|---|
| | | | | Temperature=0 | | | | |
| | EAGLE-2 | 2.77x | 2.79x | 2.87x | 2.29x | 2.27x | 3.08x | 2.68x |
| | HASS | 2.94x | 2.97x | 3.11x | 2.38x | 2.47x | 3.48x | 2.89x |
| | Gumiho | **3.04x** | 2.97x | **3.19x** | **2.58x** | **2.71x** | **3.69x** | **3.03x** |
| L3 8B | PoSS-3(E2) | 2.96x | **3.10x** | 3.17x | 2.45x | 2.50x | 3.53x | 2.95x |
| | EAGLE-3 | 2.99x | 3.11x | 3.05x | 2.34x | 2.63x | 3.62x | 2.96x |
| | EAGLE-3+HASS | 2.42x | 2.33x | 2.43x | 1.95x | 1.98x | 2.83x | 2.32x |
| | PoSS-3(E3) | **3.35x** | **3.45x** | **3.41x** | **2.71x** | **2.84x** | **3.88x** | **3.27x** |
| | EAGLE-2 | 2.99x | 2.95x | 3.23x | 2.71x | 2.49x | 3.71x | 3.01x |
| | HASS | **3.28x** | **3.34x** | 3.52x | **2.96x** | 2.72x | **4.15x** | 3.33x |
| L2 13B | Gumiho | 2.99x | 2.92x | 3.15x | 2.70x | 2.47x | 3.79x | 3.00x |
| | PoSS-3(E2) | **3.28x** | 3.32x | **3.59x** | **2.96x** | **2.74x** | 4.12x | **3.34x** |
| | | | | Temperature=1 | | | | |
| | EAGLE-2 | 2.67x | 2.55x | 2.09x | 2.02x | 2.80x | 2.47x | 2.43x |
| | HASS | **2.77x** | 2.79x | **2.14x** | 2.09x | 3.03x | 2.56x | 2.56x |
| | Gumiho | 2.40x | 2.47x | 2.55x | 2.09x | 2.07x | 2.92x | 2.42x |
| L3 8B | PoSS-3(E2) | 2.71x | **2.86x** | 2.12x | **2.18x** | **3.11x** | 2.58x | **2.59x** |
| | EAGLE-3 | 2.64x | 2.65x | 2.93x | 2.08x | 2.30x | 3.10x | 2.62x |
| | EAGLE-3+HASS | 1.80x | 1.83x | 1.91x | 1.37x | 1.57x | 2.22x | 1.79x |
| | PoSS-3(E3) | **2.90x** | **2.84x** | **3.11x** | **2.09x** | **2.49x** | **3.21x** | **2.77x** |
| | EAGLE-2 | 2.95x | 2.88x | 3.13x | 2.76x | 2.51x | 3.48x | 2.95x |
| | HASS | **3.22x** | **3.30x** | 3.46x | **2.97x** | 2.67x | 3.89x | 3.25x |
| L2 13B | Gumiho | 2.91x | 2.88x | 3.12x | 2.70x | 2.49x | 3.60x | 2.96x |
| | PoSS-3(E2) | **3.22x** | 3.23x | **3.49x** | 2.96x | **2.73x** | **3.92x** | **3.26x** |

## 4.2 MAIN RESULTS

We introduce the main results in this section. Table 1 presents the average acceptance lengths of different models. Table 2 presents the speed-up ratio of these models.

Our methods achieve the highest overall average acceptance length under different sampling temperatures, demonstrating the effectiveness of position specialists in making accurate draft predictions. In EAGLE-2 framework, **PoSS**-3(E2) achieves the best speed-up ratio under al-

Table 3: Speedup ratio under vLLM framework.

| Batch Size | 1 | 2 | 4 | 8 |
|---|---|---|---|---|
| EAGLE-3 | 1.72x | 1.79x | 1.64x | 1.67x |
| PoSS-3(E3) | **2.15x** | **2.12x** | **2.07x** | **1.87x** |

most all settings. Although Gumiho outperforms **PoSS**-3(E2) at L3 8B with temperature=0, it is less stable and performs worse under other settings. In EAGLE-3 framework, **PoSS**-3(E3) significantly outperforms the baseline on average acceptance length and the speed-up ratio. This is because the target model provides more powerful features in EAGLE-3 framework, increasing the potential of draft models to predict longer, which is what **PoSS** better at. This further demonstrates the great potential of **PoSS**: the superiority of **PoSS** over other draft methods will be greater as the input feature becomes stronger.

---

[1] https://github.com/SafeAILab/EAGLE

[2] https://github.com/HArmonizedSS/HASS

[3] https://github.com/sgl-project/SpecForge

### 4.3 POSS IN VLLM

To evaluate the performance of **POSS** on real-world application, we conduct an experiment on vLLM Kwon et al. (2023), a widely applied high-performance LLM generation framework. The experiments are conducted on A100-80GB and use Llama-3.1-8B-Instruct as the target model. We evaluate on all six datasets and take their average speedup ratio. The speedup ratio results are exhibited in Table 3. **POSS**-3(E3) consistently outperforms EAGLE-3 at different batch sizes, demonstrating the effectiveness of **POSS** on industry-standard framework.

## 5 ANALYSIS

### 5.1 POSITION-WISE ACCEPTANCE RATE

In Section 3.1, we introduce the metric **position-wise acceptance rate** (**pos-acc**) to reflect the acceptance rate of a specific position, which largely affects the overall acceptance length. Here we demonstrate that **POSS largely improves pos-acc by mitigating the feature deviation at each position and well balancing all positions**.

In Figure 4, we show the pos-acc with a draft depth of 8 on different models. EAGLE-2, with the least position generalization ability, has pos-acc lower than 65% from the $5^{th}$ position on. HASS can only maintain adequate pos-acc at the first four positions, after which performance degrades significantly due to a single draft model. EAGLE-3, with an advanced framework design, achieves higher **pos-acc** at later positions. However, the first position accuracy of EAGLE-3 drops behind other methods,

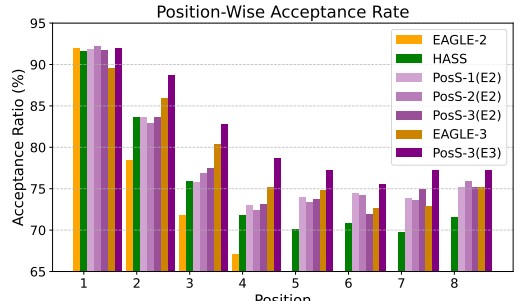

Figure 4: The position-wise acceptance rate of EAGLE, HASS, and variants of **POSS**. Experiments are conducted on MT-Bench dataset, with base model Llama-3-8B-Instruct and draft depth=8. **POSS** maintains a relatively higher pos-acc than corresponding baselines even at the $8^{th}$ position.

because the single draft model needs to balance all positions, and the **pos-acc** at the first position is sacrificed. In contrast, all variants of our **POSS** method maintain substantially higher pos-acc until the last position. The separate position specialist design also avoids the compromise of all positions. This demonstrates the effectiveness of **POSS** in mitigating position deviation and making accurate predictions.

### 5.2 COMPUTATIONAL EFFICIENCY TRADEOFF ON DRAFT DEPTH

Although tree-draft inference is widely adopted, no previous work has systematically analysed how draft depth influences generation speed. Here, we conduct a comprehensive analysis of computational costs and efficiency benefits brought by extending draft depth.

Each complete round of speculative generation involves two primary phases: the **draft phase** and the **verification phase**. In this experiment, we quantitatively analyze the time cost through three key metrics: (1) per-round computation time, (2) total round counts for test set generation, and (3) total time cost for test set generation. We demonstrate a comprehensive analysis in Figure 5 and present the following noteworthy observations.

**Larger draft depth increases draft phase computation time.** We present in Figure 5(a) the sum of per-round computation time over 5,000 rounds across varying draft depths, decomposed into draft phases and verification phases (bar chart), as well as the total rounds needed (line chart). Empirical results show that the increased total pre-round time is mainly attributed to the draft phase, and longer draft sequences do not influence verification time.

**EAGLE-3 framework reduces draft time but increases verification time.** In Figure 5(a), comparing to models of EAGLE-2 framework (HASS and **POSS**-3(E2)), models of EAGLE-3 framework (EAGLE-3 and **POSS**-3(E3)) cost less time on draft, but more time on verification. The re-

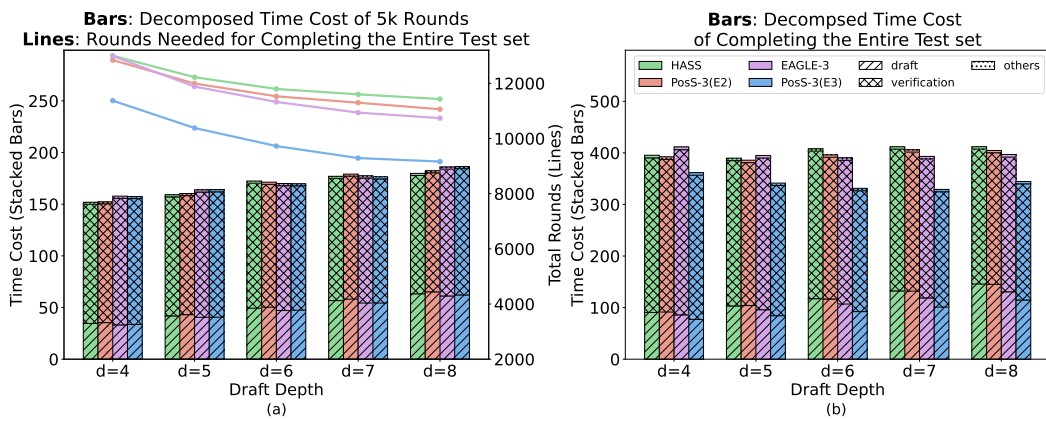

Figure 5: Computation time of different phases on MT-Bench dataset on different models across varying draft depths. The bar plots present the decomposition of time spent on each phase of speculative decoding, where subfigure (a) measures the time spent on 5k rounds and subfigure (b) measures the time to complete an entire test set. The line plot presents the number of rounds needed to complete a dataset. The lower the metrics are, the better the method is.

duction in draft time results from vocabulary-pruning setting, and the increase in verification time is because of the additional feature aggregation designs in EAGLE-3.

**POSS achieves the lowest overall computation time with reduced round counts.** The overall computation time is the multiplication of the number of rounds and the pre-round time. In Figure 5(a), the bar chart demonstrates that **POSS** has similar per-round calculation time to baseline methods, and the line chart shows that **POSS** requires fewer rounds to complete the whole test set, which is the result of a larger acceptance length. The overall time cost presented in Figure 5(b) confirms that **POSS** is faster than corresponding baselines. It is surprising that EAGLE-3 is the slowest when the draft depth is 4 and 5. This is because the first position accuracy of EAGLE-3 is negatively affected when training on large draft depth, as discussed in Section 5.1.

### 5.3 ABLATION STUDY
### ON DRAFT MODEL PREDICTION DEPTH

Figure 6 presents the throughput and average acceptance length under different draft depths. The average acceptance length $\tau$ increases with the draft depth consistently, but the improvement diminishes at higher depth. The diminishing improvement, along with the linearly increasing time cost of draft depth, creates an optimal point for throughput.

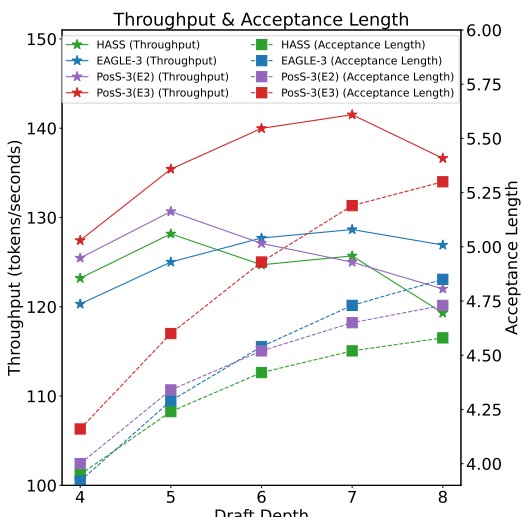

Figure 6: The throughput and average acceptance length of 4 models on different draft depths. The experiments are conducted on MT-Bench dataset. The acceptance length consistently increases as depth rises, while the throughput peaks at certain depths. This also reflects the tradeoff among different draft depths.

In the experiment on MT-Bench dataset, with Llama3-8B-Instruct as the target model, we empirically demonstrate that the throughput peaks at draft depth = 5 and 7 for models of EAGLE-2 and EAGLE-3 frameworks, respectively. This demonstrates that increasing **pos-acc** at later positions is beneficial to improving the overall throughput.

## 6 RELATED WORK

### 6.1 LINEAR SPECULATIVE DECODING

Early works (Xia et al., 2022) introduce the fundamental concept of using a draft model to predict multiple tokens in parallel. This is followed by various improvements in linear speculative decoding, including adaptive calibration techniques (Gautam et al., 2025), dynamic candidate length adjustment (Huang et al., 2024b), and methods to optimize the latency-throughput tradeoff (Sadhukhan et al., 2024). Recent advances focus on multi-token prediction (Gloeckle et al., 2024), efficient multi-sampling (Ni et al., 2024), and token recycling (Luo et al., 2024). Some also explore parallel decoding strategies with adaptive n-gram techniques (Ou et al., 2024; Wu et al., 2024; Liu et al., 2024; Wei et al., 2024).

### 6.2 TREE SPECULATIVE DECODING

Tree-based speculative decoding has advanced through several key works. GRIFFIN (Hu et al., 2025) and Sequoia (Chen et al., 2024) enhance token alignment methods, SpecInfer (Miao et al., 2024) improves sampling techniques, and Gumiho Li et al. (2025a) combines parallel and autoregressive drafting as a hybrid architecture. Other notable approaches include dynamic tree pruning (Zhong et al., 2024), early exit mechanisms (Elhoushi et al., 2024), hierarchical method (Sun et al., 2024).

### 6.3 EFFICIENT INFERENCE

Recent works apply other methods to improve the inference speed. Judge Decoding (Bachmann et al., 2025) uses a small judge model to evaluate parallel reasoning paths, while SpecReason (Pan et al., 2025) and Speculative Thinking (Yang et al., 2025) leverage speculative computation for faster inference. Other efficient reasoning techniques include efficient chain-of-thought methods (Wang et al., 2025a; Huang et al., 2025), in-context learning methods (Huang et al., 2024a), non-myopic generation (Ma et al., 2024) and system-level infra (Huang et al., 2024c).

## 7 CONCLUSION

This paper proposes **POSS**, a draft model consisting of several position specialists. This method mitigates feature deviation between the draft and target models, and reduces the deviation accumulation across draft positions. Experiments show that **POSS** maintains a high position-wise acceptance rate at later positions, achieving a larger acceptance length and faster generation speed than other methods.

## REPRODUCIBILITY STATEMENT

The experiment setup and implementation details have been disclosed in Section 4.1 and Appendix A for reproducibility. Additionally, we have carefully arranged our implementation code in an anonymous GitHub repository, `https://github.com/poss-speculative-decoding/Position-Specialist`.

## THE USE OF LARGE LANGUAGE MODELS (LLMS)

This paper uses LLM for polishing writing. Specifically, LLM is not used before the main content is written, and is only used to examine potential typos and ambiguous expressions.

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

## A IMPLEMENTATION DETAILS

### A.1 IMPLEMENTATION AT TRAINING STAGE

We mainly follow the settings of the existing work. As the implementation of EAGLE-2 and EAGLE-3 varies a lot, we separately introduce models in each framework.

Under the EAGLE-2 framework, we implement EAGLE-2, HASS, **PoSS**-1(E2), **PoSS**-2(E2), and **PoSS**-3(E2). These models are trained with the ShareGPT dataset, with 68K data entries (about 120K single dialogues after preprocessed by SpecForge (Shenggui Li, 2025)), aligning with EAGLE-1,2 and Medusa. They are trained for 40 epochs, as is implemented by HASS. All **PoSS** variants apply the losses (including loss weights) of HASS.

Under the EAGLE-3 framework, we implement EAGLE-3 and **PoSS**-3(E3). Following EAGLE-3, the UltraChat-200K dataset, with 464K data entries, is added to the training set. Despite using a much larger training set, EAGLE-3 still trains its model for 40 epochs[4]. For a fair comparison with other models, we train EAGLE-3 and **PoSS**-3(E3) for total update steps similar to models of the EAGLE-2 framework, which is 10 epochs.

In **PoSS**, the second and third layers depend on output of the previous layer. To initialize **PoSS** training, we start training from half-trained EAGLE checkpoints. In the EAGLE-2 framework, for example, the EAGLE-2 model trained for 20 epochs is used for initializing all position specialists of **PoSS**. **PoSS** is then trained for the remaining 20 epochs. In the EALGE-3 framework, this number becomes 5, and all the processes are the same.

### A.2 IMPLEMENTATION AT INFERENCE STAGE

All experiments in this paper apply the tree-draft strategy. The tree-draft inference involves three components: depth, width, and total tokens (Li et al., 2024b). As discussed in Section 5.2, a balanced depth is needed to reach the best performance. The analysis experiment results in Table 5 and Table 7 demonstrate that, for the EAGLE-2 framework, Llama3-8B-Instruct achieves the best performance on depth=6, and Llama-2-13B-chat on depth=7. The experiment result in Figure 6 suggests that models in the EAGLE-3 framework achieve the best performance in depth=7.

The influences of width and total tokens are complicated, so we apply the EAGLE-2 recommended values for them. This means the width is set to 10, and the total tokens is set to 60 for Llama3-8B-Instruct setting and 50 for Llama2-13B-chat setting.

## B DIFFERENT DRAFTING HYPERPARAMETERS

Many factors influence the average acceptance length and speed-up ratio. Besides the prediction accuracy of draft models and computational overhead, the structure of draft trees also matters. We examine two key hyperparameters that affect the performance: depth and total tokens.

We take the EAGLE-2 framework models, and conduct experiments with depths from 6 to 9. In addition to the default total tokens, we test a larger total tokens, 80. We evaluate the models on all six datasets and take the average of them. Table 4 and Table 6 present the average acceptance length. Table 5 and Table 7 present the speed-up ratio.

Interestingly, despite the consistent rise of average acceptance length as the number of total tokens increases to 80, the speed-up ratio shows a sharp drop. This indicates the target model takes significantly more time to verify. This phenomenon might result from the inner structure of the A100 GPU device that we use for experiments, which is also observed by OPT-Tree (Wang et al., 2025b).

---

[4]The number of training epochs/steps of EAGLE-3 is not disclosed in the original paper, but can be found in its official GitHub repository:https://github.com/SafeAILab/EAGLE/blob/main/eagle/traineagle3/main.py

Table 4: Average acceptance length under different hyperparameters. Experiments use Llama-3-8B-Instruct as the base model. We average the results on all six datasets. The largest average acceptance length within each column is bolded.

| Temperature | Depth | 6 | | 7 | | 8 | | 9 | |
|---|---|---|---|---|---|---|---|---|---|
| | Total Tokens | 60 | 80 | 60 | 80 | 60 | 80 | 60 | 80 |
| T=0 | HASS | 4.39 | 4.49 | 4.49 | 4.62 | 4.54 | 4.67 | 4.59 | 4.73 |
| | PosS-1(E2) | 4.54 | 4.64 | 4.65 | 4.78 | **4.74** | 4.89 | **4.79** | 4.94 |
| | PosS-2(E2) | **4.55** | **4.67** | **4.68** | **4.81** | **4.74** | **4.90** | **4.79** | **4.96** |
| | PosS-3(E2) | 4.50 | 4.62 | 4.61 | 4.75 | 4.69 | 4.83 | 4.73 | 4.89 |
| T=1 | HASS | 4.16 | 4.24 | 4.22 | 4.34 | 4.26 | 4.39 | 4.30 | 4.41 |
| | PosS-1(E2) | **4.28** | **4.37** | 4.35 | 4.48 | **4.44** | **4.58** | 4.47 | 4.58 |
| | PosS-2(E2) | 4.27 | **4.37** | **4.37** | **4.53** | 4.43 | 4.57 | **4.48** | **4.64** |
| | PosS-3(E2) | **4.28** | 4.35 | 4.30 | 4.49 | 4.40 | 4.53 | 4.43 | 4.53 |

Table 5: Speed-up ratio under different hyperparameters. Experiments use Llama-3-8B-Instruct as the base model. We average the results on all six datasets. The largest number within each row is bolded to show the upper bound of each method.

| Temperature | Depth | 6 | | 7 | | 8 | | 9 | |
|---|---|---|---|---|---|---|---|---|---|
| | Total Tokens | 60 | 80 | 60 | 80 | 60 | 80 | 60 | 80 |
| T=0 | HASS | **2.89x** | 2.83x | 2.84x | 2.78x | 2.76x | 2.71x | 2.67x | 2.65x |
| | PosS-1(E2) | **2.94x** | 2.90x | 2.90x | 2.85x | 2.83x | 2.80x | 2.76x | 2.72x |
| | PosS-2(E2) | **2.98x** | 2.92x | 2.93x | 2.87x | 2.84x | 2.81x | 2.77x | 2.74x |
| | PosS-3(E2) | **2.95x** | 2.89x | 2.89x | 2.84x | 2.83x | 2.78x | 2.73x | 2.71x |
| T=1 | HASS | **2.63x** | 2.54x | 2.56x | 2.50x | 2.47x | 2.44x | 2.41x | 2.35x |
| | PosS-1(E2) | **2.73x** | 2.65x | 2.66x | 2.59x | 2.60x | 2.55x | 2.53x | 2.48x |
| | PosS-2(E2) | **2.66x** | 2.60x | 2.63x | 2.57x | 2.55x | 2.51x | 2.48x | 2.45x |
| | PosS-3(E2) | **2.67x** | 2.59x | 2.60x | 2.56x | 2.55x | 2.47x | 2.48x | 2.41x |

## C   EXTRA MEMORY USAGE DURING INFERENCE

Involving extra draft layers requires extra GPU memory usage, and the GPU memory usage increases linearly with the number of position specialists. Fortunately, this additional cost is negligible compared to the target model size since each specialist costs only one transformer layer (around 218M parameters per specialist for an 8B target model).

Empirically, Figure 7 visualizes the memory usage of the single draft model and **PosS**-1,2,3. Here, EAGLE and HASS cost the same GPU memory, and they are de facto **PosS**-∞. Assuming the draft depth is 6, the draft layers in the methods are 1, 2, 3, and 6, from left to right. In both target model settings, **PosS**-3 and **PosS**-2 increase a few extra memory usage. **PosS**-1, despite using 6 times draft layers than EAGLE-2, costs acceptable extra memory usage.

## D   DYNAMIC LAYER ALLOCATION

Throughout this paper, the degree of specialization, i.e., the number of positions allocated to a layer, is fixed to 1, 2, or 3. In this section, we discuss the dynamic design of layer allocation.

A straightforward way to allocate multiple layers is Mixture-of-Experts (MoE). We design **PosS**-3(E3)-MoE to investigate if standard MoE works for **PosS** or not. Specifically, **PosS**-3(E3)-MoE applies the structure of **PosS**-3(E3), and a light-weight router to dynamically decide which layer to use at each position. The router functions in the same manner during training and inference. **PosS**-3(E3)-MoE uses the same training data and training steps as **PosS**-3(E3). The complete evaluation results on all six datasets are exhibited in Table 8.

Table 6: Average acceptance length under different hyperparameters. Experiments use Llama-2-13B-chat as the base model. We average the results on all six datasets. The largest average acceptance length within each column is bolded.

| Temperature | Depth | 6 | | 7 | | 8 | | 9 | |
|---|---|---|---|---|---|---|---|---|---|
| | Total Tokens | 50 | 80 | 50 | 80 | 50 | 80 | 50 | 80 |
| T=0 | HASS | 4.68 | 5.20 | 5.21 | 5.45 | 5.46 | 5.62 | 5.57 | 5.75 |
| | PosS-1(E2) | 5.09 | 5.20 | 5.24 | 5.48 | 5.52 | 5.66 | 5.63 | 5.79 |
| | PosS-2(E2) | **5.13** | **5.22** | 5.25 | 5.49 | 5.53 | 5.68 | 5.65 | 5.82 |
| | PosS-3(E2) | **5.13** | 5.21 | **5.27** | **5.51** | **5.55** | **5.70** | **5.66** | **5.83** |
| T=1 | HASS | **4.90** | 5.06 | 5.03 | 5.29 | 5.24 | 5.45 | 5.35 | 5.52 |
| | PosS-1(E2) | 4.89 | **5.11** | **5.13** | 5.31 | **5.34** | 5.49 | 5.43 | 5.52 |
| | PosS-2(E2) | 4.87 | **5.11** | 5.03 | **5.32** | 5.30 | 5.49 | **5.44** | 5.61 |
| | PosS-3(E2) | 4.89 | **5.11** | 5.09 | 5.31 | 5.33 | **5.50** | 5.43 | **5.62** |

Table 7: Speed-up ratio under different hyperparameters. Experiments use Llama-2-13B-chat as the base model. We average the results on all six datasets. The largest number within each row is bolded to show the upper bound of each method.

| Temperature | Depth | 6 | | 7 | | 8 | | 9 | |
|---|---|---|---|---|---|---|---|---|---|
| | Total Tokens | 50 | 80 | 50 | 80 | 50 | 80 | 50 | 80 |
| T=0 | HASS | 3.28x | 3.02x | **3.33x** | 3.08x | 3.31x | 3.09x | 3.28x | 3.09x |
| | PosS-1(E2) | 3.16x | 2.93x | **3.21x** | 3.08x | **3.21x** | 3.09x | 3.20x | 3.09x |
| | PosS-2(E2) | 3.26x | 3.00x | 3.30x | 3.06x | **3.31x** | 3.09x | 3.27x | 3.07x |
| | PosS-3(E2) | 3.29x | 3.00x | 3.34x | 3.09x | **3.35x** | 3.11x | 3.30x | 3.10x |
| T=1 | HASS | 3.24x | 2.94x | **3.25x** | 3.00x | 3.20x | 3.01x | 3.18x | 2.98x |
| | PosS-1(E2) | 3.13x | 2.93x | **3.17x** | 2.95x | 3.14x | 2.97x | 3.10x | 2.92x |
| | PosS-2(E2) | 3.17x | 2.94x | **3.19x** | 2.98x | 3.18x | 2.99x | 3.17x | 2.98x |
| | PosS-3(E2) | 3.24x | 2.97x | **3.26x** | 3.00x | **3.26x** | 3.02x | 3.18x | 3.01x |

The results clearly show that standard MoE does not work with **PosS**, and here is an explanation for it. In standard MoE, all experts are counterparts of each other, whose input and output are in the same hidden state space. In **PosS**, however, each subsequent layer refines the feature bias produced by the preceding one. Therefore, the input and output spaces are different for each position specialist, and randomly mixing them introduces noise that confuses the model. This result, on the other hand, proves the necessity of layer specialization.

Nevertheless, dynamic layer allocation is still a promising direction as long as it preserves the sequential order of layers. Equation 3 reveals that the accuracy at one position influences the acceptance rate of all following positions, highlighting the importance of correctly predicting the first few positions. Therefore, it should be beneficial to let the first layer take charge of fewer positions and assign more positions to later layers. For example, we can change the (3,3) allocation strategy of **PosS**-3 into (2,4). The allocation becomes more complex when draft depth increases, where a trainable module might be helpful. We leave this exploration for future work.

Table 8: Experiments of fixed layer allocation and MoE-based layer allocation. The fixed allocation method performs better in terms of average acceptance length and speedup ratio, proving that layer specialization is necessary for **PosS**, and mixing the order of layers causes performance degradation.

| Model | MT-Bench | | Alpaca | | GSM8K | | Natural Questions | | CNN/DM | | HumanEval | | Avg. | |
|---|---|---|---|---|---|---|---|---|---|---|---|---|---|---|
| | $\tau$ | speedup | $\tau$ | speedup | $\tau$ | speedup | $\tau$ | speedup | $\tau$ | speedup | $\tau$ | speedup | $\tau$ | speedup |
| Temperature=0 | | | | | | | | | | | | | | |
| PosS-3(E3) | **5.15** | **3.35x** | **5.50** | **3.45x** | **5.43** | **3.41x** | **4.13** | **2.71x** | **4.54** | **2.84x** | **5.95** | **3.88x** | **5.12** | **3.27x** |
| PosS-3(E3)-MoE | 4.52 | 2.88x | 4.72 | 2.95x | 5.02 | 3.13x | 3.87 | 2.45x | 3.98 | 2.53x | 5.43 | 3.51x | 4.62 | 2.91x |
| Temperature=1 | | | | | | | | | | | | | | |
| PosS-3(E3) | **4.66** | **2.90x** | **4.98** | **2.84x** | **5.24** | **3.11x** | **3.81** | **2.09x** | **4.14** | **2.49x** | **5.69** | **3.21x** | **4.75** | **2.77x** |
| PosS-3(E3)-MoE | 3.53 | 1.86x | 4.12 | 2.18x | 4.22 | 2.14x | 3.17 | 1.72x | 3.55 | 1.84x | 5.00 | 2.52x | 3.93 | 2.05x |

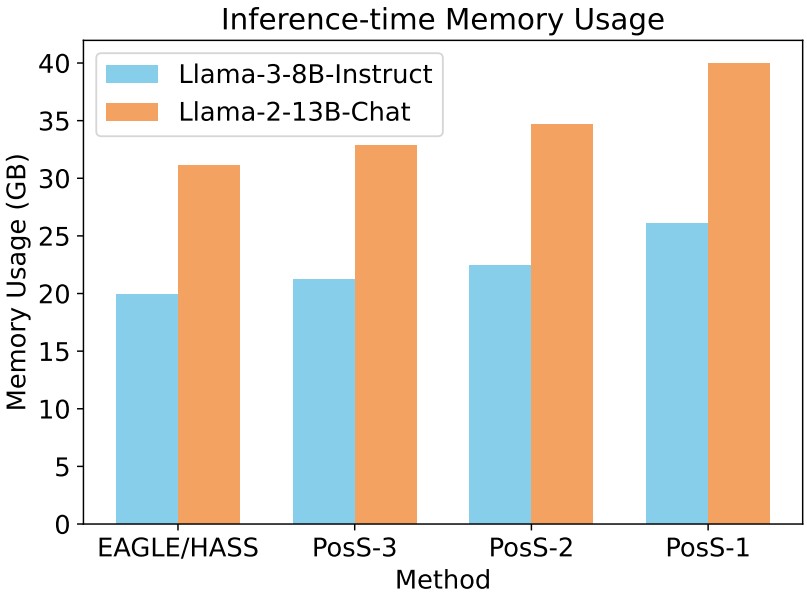

Figure 7: The Inference-time GPU memory usage of different speculative decoding methods. The memory usage is measured on the MT-bench test dataset. **POSS** methods require slightly more GPU memory than EAGLE, the baseline method.

