# OpenReview forum: "PosS: Position Specialist Generates Better Draft for Speculative Decoding"
_ICLR.cc/2026/Conference — Submitted to ICLR 2026_

### Official Review · Reviewer_udoP · 2025-10-15

**Soundness:** 3
**Presentation:** 3
**Contribution:** 2
**Rating:** 4
**Confidence:** 4

**Summary:**

This paper addresses error accumulation in speculative decoding, a phenomenon where the quality of draft tokens progressively degrades at later positions in a sequence. To mitigate this, they introduce Position Specialists (PosS), a novel approach that replaces a single draft model with multiple, position-specialized draft layers. Each specialist is explicitly trained to predict the token for a specific set of positions. Extensive experiments on Llama-2-13B and Llama-3-8B models across six datasets demonstrate that PosS outperforms baseline methods like EAGLE-3, achieving an approximately 10% higher speed-up ratio.

**Strengths:**

1. Clear Motivation: The paper clearly identifies the degradation of draft quality over sequence length. The proposed solution of using specialized models for different positions is intuitive.
2. Useful Metric: The introduction of the "position-wise acceptance rate" (pos-acc) is a valuable contribution.
3. Strong Baseline: The inclusion of strong baselines like EAGLE-3 and HASS makes the comparison reasonable.

**Weaknesses:**

1. Missing Comparison to Parameter-Matched Models: The paper fails to compare PosS against a single draft model with an equivalent parameter count (e.g., using an MoE layer, which makes the single draft model become multiple experts, and can also route between different experts dynamically instead of a pre-defined order in PosS). This makes it impossible to determine if the benefits come from "specialization" or simply from increased model capacity.
2. High Deployment Complexity: Maintaining independent KV-Caches for each specialist and switching between them introduces significant overhead. This complexity poses challenges for deployment, especially within modern serving frameworks.
3. Unfavorable Cost-Benefit Ratio: The performance gains are often marginal and inconsistent, as seen in Table-2, the L2-13B model, where PosS (3.34x) offers negligible improvement over HASS (3.33x). Such limited benefits do not justify the substantial increase in system complexity and overhead.

**Questions:**

1. In the appendix, the ablation studies in Tables 3 and 4 show that PosS-2 almost always outperforms PosS-3. Why were the experiments in the main text conducted with PosS-3 instead?

---

> ### Author Response · Authors · 2025-11-21
> **Response to Reviewer udoP**
>
> Thanks for your insightful comments. We provide a point-by-point response to all your questions here. We sincerely hope our responses address your concerns, and hope you can reconsider your assessment of our work if you find our response satisfactory.
>
> ---
>
> **W1**: Missing Comparison to Parameter-Matched Models: The paper fails to compare PosS against a single draft model with an equivalent parameter count (e.g., using an MoE layer, which makes the single draft model become multiple experts, and can also route between different experts dynamically instead of a pre-defined order in PosS). This makes it impossible to determine if the benefits come from "specialization" or simply from increased model capacity.
>
> **A1**: Thanks for your insightful suggestion, and we design an MoE-based method, named *PosS-3(E3)-MoE*, accordingly to compare with PosS. Specifically, we keep the multi-layer model structure of PosS, but **add a router to dynamically decide** which layer to use for each position. The table below shows the comparison between *PosS-3(E3)* and *PosS-3(E3)-MoE*, and we also report the complete results in Table 8 in our revised paper.
>
> |  | Temperature=0 |  | Temperature=1 |  |
> |---|:---:|:---:|:---:|:---:|
> |  | $\tau$ | Speedup | $\tau$ | Speedup |
> | PosS-3(E3) | **5.12** | **3.27x** | **4.75** | **2.77x** |
> | PosS-3(E3)-MoE | 4.62 | 2.91x | 3.93 | 2.05x |
>
> As shown in the table above (and Table 8 in our revised paper), the MoE version significantly underperformed our PosS method. This proves that simply increasing capacity (parameters) is insufficient. The performance gain of PosS mainly comes from its **sequential specialization**, where subsequent layers are trained to refine the feature deviation of previous layers.
>
> We have added the discussion to Appendix D in our revised paper.
>
> ---
>
> **W2**: High Deployment Complexity: Maintaining independent KV-Caches for each specialist and switching between them introduces significant overhead. This complexity poses challenges for deployment, especially within modern serving frameworks.
>
> **A2**: We clarify that the introduced complexity and overhead are limited.
>
> **Memory**: Figure 7 illustrates the GPU memory usage of PosS and EAGLE/HASS. On different models, PosS-3 consumes only slightly extra GPU memory than EAGLE/HASS.
>
> **Latency**: Figure 5 decomposes the inference time cost into draft and verification, which clearly shows that PosS increases the draft time cost by only a margin. In addition, we deployed PosS-3(E3) on vLLM, which consistently outperforms EAGLE-3 under all batch sizes. The experiment results on vLLM are listed in the table below. We also added these results to Table 3 of our revised paper.
>
> **Speedup ratio on vLLM**:
> | Batch Size | 1 | 2 | 4 | 8 |
> |---|:---:|:---:|:---:|:---:|
> | EAGLE-3 | 1.72x | 1.79x | 1.64x | 1.67x |
> | PosS-3(E3) | **2.15x** | **2.12x** | **2.07x** | **1.87x** |
>
> This further demonstrates the effectiveness of PosS in real-world deployment.
>
> ---
>
> **W3**: Unfavorable Cost-Benefit Ratio: The performance gains are often marginal and inconsistent, as seen in Table 2, the L2-13B model, where PosS (3.34x) offers negligible improvement over HASS (3.33x). Such limited benefits do not justify the substantial increase in system complexity and overhead.
>
> **A3**: While the gap on Llama-2-13B is smaller, the gap on the stronger EAGLE-3 is larger (PosS 3.27x over EAGLE-3 2.96x), and our PosS consistently outperforms all baselines under different settings.
>
> **The trend is the key**: PosS scales better to feature-enriched target models like EAGLE-3 architecture (which has low-, mid-, and high-level features).  This is because PosS has greater potential in preserving a high position-wise accuracy when the target model provides a stronger feature that can lead to more accurate draft tokens.  This is proven by experiment results on EAGLE-3, where PosS-3(E3) significantly outperforms EAGLE-3 baseline, in both standard inference settings and vLLM deployment. Given the industry shift toward stronger models, PosS has a future-promising advantage.
>
> ---
>
> **Q1**: In the appendix, the ablation studies in Tables 3 and 4 show that PosS-2 almost always outperforms PosS-3. Why were the experiments in the main text conducted with PosS-3 instead?
>
> **A4**:
> We selected PosS-3 for two reasons:
>
> **Stability**: We clarify that PosS-2 does not always outperform PosS-3: PosS-2 performs better at temperature=0, but worse at temperature=1. In fact, PosS-2 is the slowest at temperature=1. On the contrary, PosS-3 is the most stable setting across different temperatures and target models.
>
> **Memory-Efficiency**: Beyond the empirical experiment results, we select PosS-3 because it involves fewer specialist layers, and thus less overhead of computation and memory cost.

---

### Official Review · Reviewer_PRFX · 2025-10-28

**Soundness:** 3
**Presentation:** 3
**Contribution:** 2
**Rating:** 4
**Confidence:** 5

**Summary:**

This paper propose Position Specialists(PosS), which consist of multiple position-specialized draft layers to generate tokens at assigned position. They also introduce position-wise to measure the conditional probability of accepting the i-th token given the acceptance of its preceding (i-1)-th token. Experimental results show the effectiveness of the proposed method.

**Strengths:**

1. This paper is technically sound and easy to understand.
2. The experimental results show the effectiveness of the proposed method.

**Weaknesses:**

1. What is the difference between PosS and Medusa, which also uses different heads for different token positions? It seems that PosS-1 is exactly the same as Medusa.
2. Eq.4 seems degenerate to P(A_k),  why don’t you describe Eq.4 like that?
3. What is the performance when changing the architecture of PosS-1 to Medusa and leaving other things unchanged?

**Questions:**

See weaknesses above. The difference to Medusa is important.

---

> ### Author Response · Authors · 2025-11-21
> **Response to Reviewer PRFX**
>
> Thanks for your comments. We provide a point-by-point response to all your questions here. We sincerely hope our responses address your concerns, and hope you can reconsider your assessment of our work if you find our response satisfactory.
>
> ---
>
> **W1**: What is the difference between PosS and Medusa, which also uses different heads for different token positions? It seems that PosS-1 is exactly the same as Medusa.
>
> **A1**:
> PosS and Medusa are fundamentally different methods:
>
> (1) The key difference between PosS and Medusa lies in the order of generating tokens. While Medusa generates all tokens **in parallel**, PosS generates them **autoregressively**. Although the autoregressive generation consumes more time, the generated drafts are significantly more accurate, leading to a better speedup ratio.
>
> (2) PosS and Medusa are also different in model structure. Medusa uses MLPs (the Medusa Head) to generate draft tokens **independently**. PosS uses one-layer transformers to conduct **context-conditioned** draft tokens prediction.
>
> ---
>
> **W2**: Eq.4 seems degenerate to P(A_k), why don’t you describe Eq.4 like that?
>
> **A2**: You are right that $P(A_1 \cap A_2,..., A_k) = P(A_k)$. Eq.4 decomposes $P(A_k)$ into chain rules mainly to show the multiplicative degradation of speculative decoding. Therefore, the prediction accuracy at the beginning positions is more important than those at the end positions.
> We add $P(A_k)= P(A_1 \cap A_2,..., A_k)$ to Eq.4, and have updated it in our revised paper.
>
> ---
>
> **W3**: What is the performance when changing the architecture of PosS-1 to Medusa and leaving other things unchanged?
>
> **A3**: It is impossible to change the architecture of PosS-1 (one-layer Transformer) to Medusa (MLP) without leaving other things unchanged. The core algorithm of EAGLE-styled method (including PosS and HASS) is to generate draft tokens with contexts:
> $P(x_{t+k}|x_{<t+k},f_{<t+k-1})$
> , compared to Medusa:
> $P(x_{t+k}|f_t)$
> , where $x$ denotes token and $f$ denotes feature.
>
> MLP is sufficient for the context-free method (Medusa), but is unable in modeling the context-conditioned method (PosS-1). Therefore, the proposition that *changing the architecture of PosS-1 to Medusa and leaving other things unchanged* is invalid.
>
> We would like to provide experimental results to further address your concern about the difference of Medusa and PosS. As shown below, the context-aware PosS approach is far superior to the Medusa approach.
>
> **Acceptance length**:
> |        | **MT_Bench** | **Alpaca** | **GSM8K** | **Natural Questions** | **CNN/DM** | **HumanEval** | **Avg.** |
> |:------:|:------------:|:----------:|:---------:|:---------------------:|:----------:|:-------------:|:--------:|
> | Medusa |     2.44     |    2.49    |    2.47   |          2.44         |    2.45    |      2.42     |   2.45   |
> | PosS-1 |     **4.54**     |    4.78    |    **4.82**   |          **3.65**         |    **4.06**    |      5.39     |   **4.54**   |
> | PosS-3 |     4.52     |    **4.82**    |    4.81   |          3.64         |    4.05    |      **5.41**     |   **4.54**   |
>
> **Speedup ratio**:
> |        | **MT_Bench** | **Alpaca** | **GSM8K** | **Natural Questions** | **CNN/DM** | **HumanEval** | **Avg.** |
> |:------:|:------------:|:----------:|:---------:|:---------------------:|:----------:|:-------------:|:--------:|
> | Medusa |     2.03x    |    2.17x   |   2.13x   |         2.08x         |    2.10x   |     2.08x     |   2.10x  |
> | PosS-1 |     **2.96x**    |    3.00x   |   **3.19x**   |         **2.49x**         |    **2.50x**   |     3.52x     |   2.94x  |
> | PosS-3 |     **2.96x**    |    **3.10x**   |   3.17x   |         2.45x         |    **2.50x**   |     **3.53x**     |   **2.95x**  |

---

### Official Review · Reviewer_dNHR · 2025-10-31

**Soundness:** 3
**Presentation:** 4
**Contribution:** 3
**Rating:** 4
**Confidence:** 5

**Summary:**

This paper introduces Position Specialists (PoSS), a novel architectural framework designed to enhance the accuracy and efficiency of speculative decoding for Large Language Models. The core problem addressed is the rapid degradation of draft prediction quality at later positions, which is caused by the accumulation of feature deviation between the draft model's generated features and the target model's features. PoSS resolves this by replacing the single draft layer with multiple position-specialized draft layers, each trained to handle the expected level of feature deviation at its assigned position range.

**Strengths:**

- The introduction of the position-wise acceptance rate (pos-acc) metric provides a crucial analytical tool for diagnosing and comparing the efficiency of different speculative decoding methods at a granular level.

- The Position Specialists (PoSS) architecture is a novel and intuitive approach that effectively mitigates the fundamental challenge of accumulated feature deviation by distributing the prediction task across multiple specialized draft layers.

**Weaknesses:**

1. The performance gain achieved by the proposed method is difficult to solely attribute to the "Position Specialists" architecture, as opposed to the "HASS-like" recursive feature alignment training strategy (which PoSS utilizes). Specifically, the incremental improvement of PosS-3(E2) over the HASS baseline is marginal (an average of only 2.1% on L3-8B and 0.3% on L2-13B, based on the speedup ratio from Table 2 at t=0). This small margin suggests that the performance lift might primarily stem from the training approach common to both HASS and PosS.

2. This work shares conceptual similarity with an existing approach[1] which also emphasizes the differentiated treatment of draft tokens at different positions. A clear discussion detailing the differences and experimental results between the proposed POSS method and the aforementioned work is better.

[1] Gumiho: A Hybrid Architecture to Prioritize Early Tokens in Speculative Decoding

**Questions:**

- Following weakness 1, to definitively demonstrate the effectiveness of the proposed Position Specialists architecture, the authors should design an ablation study that isolates its contribution from the HASS training method. A critical experiment would involve training the EAGLE-3 architecture (i.e., the single draft layer) using the same training data and total steps as PosS-3(E3), but without the position specialists. Comparing this result directly against PosS-3(E3) would clearly indicate the incremental value of the PoSS architecture itself. If you solve this core problem, I'm willing to raise my score.

- How is the Key-Value (KV) cache managed under the PoSS framework? Since different specialist layers  are used to predict tokens at different positions, the features generated by these layers are distinct. This prevents the standard sharing and reusing of KV caches that is typical in single-model decoding. Could the authors quantify the potential overhead introduced by managing non-shared KV caches and discuss whether this processing time significantly impacts the overall speedup?

- The authors should provide performance results using industry-standard, high-performance serving frameworks such as vllm or sglang. Providing this data would significantly increase the practical relevance and utility of the proposed method for real-world deployment.

---

> ### Author Response · Authors · 2025-11-20
> **Response to Reviewer dNHR (1/2)**
>
> Thanks for your valuable comments. We provide a point-by-point response to all your questions here. We sincerely hope our responses address your concerns, and hope you can reconsider your assessment of our work if you find our response satisfactory.
>
> ---
>
> **W1 & Q1**: The performance gain achieved by the proposed method is difficult to solely attribute to the "Position Specialists" architecture, as opposed to the "HASS-like" recursive feature alignment training strategy (which PoSS utilizes). To definitively demonstrate the effectiveness of the proposed Position Specialists architecture, the authors should design an ablation study that isolates its contribution from the HASS training method. A critical experiment would involve training the EAGLE-3 architecture (i.e., the single draft layer) using the same training data and total steps as PosS-3(E3), but without the position specialists. Comparing this result directly against PosS-3(E3) would clearly indicate the incremental value of the PoSS architecture itself. If you solve this core problem, I'm willing to raise my score.
>
> **A1**: We would like to clarify that the recursive feature alignment training strategy is applied by HASS, EAGLE-3, and PosS. **The comparison of EAGLE-3 and PosS-3(E3) demonstrates that the improvement of PosS over EAGLE-3 is not derived from the recursive training.**
>
> Below, we compare the training loss(es) of these methods:
>
> * **EAGLE-3**: cross-entropy loss between draft and target token
> * **PosS(E3)**: cross-entropy loss between draft and target token
> * **HASS**:  cross-entropy loss between draft and target token + regression loss between draft and target hidden representation + topk distillation loss
>
> We conduct experiments using EAGLE-3 architecture with HASS’s training loss as below (without position specialists), and report the experiment results in the table below, which shows the average evaluation on all six datasets. We also reported the full results on six datasets in our updated Table 1 and 2.
> |  | Temperature=0 |  | Temperature=1 | |
> |---|:---:|:---:|:---:|:---:|
> |  | $\tau$ | Speedup | $\tau$ | Speedup |
> | EAGLE-3 | 4.69 | 2.96x | 4.38 | 2.62x |
> | EAGLE-3+HASS | 3.71 | 2.32x | 3.28 | 1.79x |
> | PosS-3(E3) | 5.12 | 3.27x | 4.75 | 2.77x |
>
>
> As shown in this table (and Tables 1 & 2 in our revised paper), the HASS training strategy actually hurts the EAGLE-3 model performance (it imposes additional constraints that are inconsistent with the ultimate goal of token prediction), while our PosS method outperforms all the baselines.
>
> We thank you for this insightful suggestion, and we have supplemented this method setting to Tables 1 and 2 in our revised paper to better demonstrate the effectiveness of PosS.
>
> ---
>
> **W2**: This work shares conceptual similarity with an existing approach[1] which also emphasizes the differentiated treatment of draft tokens at different positions. A clear discussion detailing the differences and experimental results between the proposed POSS method and the aforementioned work is better.
>
> **A2**: We have added a detailed comparison in experiment and related work sections. The approach Gumiho [1] shares some conceptual similarity with PosS, yet PosS outperforms Gumiho conceptually and empirically. Gumiho mixes auto-regressive and parallel drafting: generation of the first two draft positions is auto-regressive (EAGLE-like), and the following positions are generated in parallel (Medusa-like).
>
> (1) Conceptually, parallel drafting predicts less accurate tokens than auto-regressive drafting, which has been proven by EAGLE. What is worse, tokens generated in parallel (Medusa-like) have combinations of an exponential number, demanding a heavy verification load for the target model. PosS, however, consistently predicts accurate drafts while keeping verification load at a low level.
>
> (2) Empirically, we reproduce Gumiho with their open-sourced code and checkpoint, and the results are shown below.
>
> |        |         | Temperature=0 |           | Temperature=1 |           |
> |:------:|:-------:|:-------------:|:---------:|:-------------:|:---------:|
> |  Model |  Method | $\tau$ |  Speedup  | $\tau$ |  Speedup  |
> |  L3 8B | EAGLE-2 |      4.06     |   2.68x   |      3.86     |   2.43x   |
> |        |  Gumiho |      4.28     | **3.03x** |      3.99     |   2.42x   |
> |        |  PosS-3(E2) |    **4.54**   |   2.95x   |    **4.25**   | **2.59x** |
> | L2 13B | EAGLE-2 |      4.79     |   3.01x   |      4.64     |   2.95x   |
> |        |  Gumiho |      4.78     |   3.00x   |      4.62     |   2.96x   |
> |        |  PosS-3(E2) |    **5.27**   | **3.34x** |    **5.09**   | **3.26x** |
>
> We have added a citation to this related work and the complete results of Gumiho to our revised paper.
>
> [1] Gumiho: A Hybrid Architecture to Prioritize Early Tokens in Speculative Decoding

---

> ### Author Response · Authors · 2025-11-20
> **Response to Reviewer dNHR (2/2)**
>
> **Q2**: How is the Key-Value (KV) cache managed under the PoSS framework? Since different specialist layers are used to predict tokens at different positions, the features generated by these layers are distinct. This prevents the standard sharing and reusing of KV caches that is typical in single-model decoding. Could the authors quantify the potential overhead introduced by managing non-shared KV caches and discuss whether this processing time significantly impacts the overall speedup?
>
> **A3**: This is an insightful perspective, and we clarify that the modified KV cache brings only **slight impacts** on the processing time. In PosS, every specialist layer stores an independent KV cache, and all forwards using the same layer share the KV cache. Considering PosS-3 with draft depth 6, the number of layers is equal to 2. In this case, we maintain one more separate KV cache than the standard Transformer.
>
> Next, we roughly calculate the **FLOPs at the draft stage** of EAGLE and PosS-3 to compare the additional computation cost.
>
> Let $L$ denote sequence length and $d$ denote model dimension.
> * **EAGLE**:
>   * The projection of Q, K, V, O costs $2Ld^2$ each => $8Ld^2$ in total.
>   * The multiplication of $QK^T$ and its multiplication with $V$ cost $2L^2d$ each => $4L^2d$ in total.
>   * MLP: it contains two matrix multiplications, where the FFN dimension is $3.5d$ for Llama3-8B. The computation cost of each multiplication is $2Ld*3.5d=7Ld^2$ => $14Ld^2$ in total.
>   * Overall FLOPs: $22Ld^2 + 4L^2d$
>
> * **PosS-3**:
>   * Everything is the same as **EAGLE**, except for the extra projection of uncached $K$ and $V$, each costing $2Ld^2$ => $4Ld^2$ in total
>
> Now, we can calculate the ratio of additional computation cost at **draft stage**: $4Ld^2/(22Ld^2 + 4L^2d) < 4Ld^2/22Ld^2 = 18.18$%
>
> Eventually, we calculate the ratio of additional computation cost in **a complete draft-verification round**.
>
> As the target model has 32 Transformer layers and the draft model has 1 Transformer layer, the additional computation cost is about $18.18$%$ \times 1/32=0.57$% of the entire draft-verification round. This proves that the potential overhead introduced is marginal.
>
> In addition, Figure 5(a) provides the empirical statistics of time cost at the draft phase. Comparing either HASS and PosS(E2) or EAGLE-3 and PosS(E3), we can see the empirical additional time cost is quite marginal.
>
> ---
>
> **Q3**: The authors should provide performance results using industry-standard, high-performance serving frameworks such as vllm or sglang. Providing this data would significantly increase the practical relevance and utility of the proposed method for real-world deployment.
>
> **A4**: We appreciate this valuable suggestion, and implemented PosS on the vLLM framework. The average speedup ratio on six datasets of PosS-3(E3) and EAGLE-3 is listed in the table below, demonstrating the effectiveness of PosS-3(E3) under real-world deployment.
> | Batch Size | 1 | 2 | 4 | 8 |
> |:---:|:---:|:---:|:---:|:---:|
> | EAGLE-3 | 1.72x | 1.79x | 1.64x | 1.67x |
> | PosS-3(E3) | **2.15x** | **2.12x** | **2.07x** | **1.87x** |
>
> **We have also added this result to Section 4.3 of our revised paper.**

---

> ### Comment · Reviewer_dNHR · 2025-11-24
>
> Thank you for your valuable response, which has thoroughly addressed all my questions. I am now willing to raise my score.
>
> However, I have one final question regarding the new ablation study presented in your response to W1/Q1.
>
> Specifically: The new result shows that EAGLE-3 + HASS performs significantly worse than the standalone EAGLE-3 model. This contradicts the observation that HASS improves performance when applied to the EAGLE-2 framework. Could the authors please provide an analysis of this phenomenon? Why does the addition of the HASS training loss hurt the performance of the EAGLE-3 architecture?
>
> More broadly, why does the HASS approach (which showed a significant improvement over EAGLE-2) fail to improve, and instead degrade, performance when applied to the EAGLE-3 architecture?
>
> I appreciate your further clarification on this point.

---

> > ### Author Response · Authors · 2025-11-24
> >
> > We sincerely appreciate your recognition of our response. We respond to your follow-up question in the following content.
> >
> > HASS training loss hurts the performance of EAGLE-3 architecture because of its feature loss. We direct you to the 2nd page of EAGLE-3 [1], where it points out that the feature loss may harm EAGLE-3 architecture, though benefiting EAGLE-1/2’s architecture:
> >
> > “*EAGLE’s loss function consists of two components: the feature prediction loss $l_{fea}$ and the token prediction loss $l_{token}$. Thanks to the feature prediction loss, the draft model trained only at Step 1 can adapt to Step 2 and acquire multi-step prediction capabilities. However, **with token prediction as the ultimate goal, feature prediction can be seen as an additional constraint, which limits the expressiveness of the draft model and makes it difficult to benefit from increased data**.*”
> >
> > Conceptually, our understanding is that in the EAGLE-3 architecture, the feature loss needs to align the combination of low-, mid-, and high-level features of the target model, and this optimization becomes significantly harder (compared to only aligning the last-layer hidden representation in EAGLE-2). This explains why HASS fails to improve EAGLE-3 architecture.
> >
> > We sincerely hope our response can address your concern.
> >
> > [1] EAGLE-3: Scaling up Inference Acceleration of Large Language Models via Training-Time Test

---

### Official Review · Reviewer_ATMa · 2025-11-01

**Soundness:** 3
**Presentation:** 3
**Contribution:** 3
**Rating:** 6
**Confidence:** 3

**Summary:**

This paper introduces a new framework named Position Specialists (PosS) to address the issue of degrading acceptance rates as the speculative step increases. The authors introduce a novel metric "pos-acc" to quantify and analyze this problem. The proposed method demonstrates significant improvements in both average acceptance length and speed-up ratios over strong baselines.

**Strengths:**

1. The core motivation is clear and compelling. The paper accurately identifies a critical and practical problem in existing methods: the "accumulated feature deviation" and the limited generalization capability of a single draft model. The paper is well-written, with a logical flow and clear exposition.
2. The design of PosS is elegant and intuitive, employing different "specialist" models to handle tasks of varying difficulty (i.e., different levels of feature deviation). The effectiveness of the approach is strongly supported by comprehensive experiments.

**Weaknesses:**

1. The paper mentions "tree-draft" in Section 6.2 but does not sufficiently clarify its relationship with the experiments. It's unclear whether "draft depth" in Sections 6.2 and 6.3 refers to the depth of the tree or simply the length of a linear draft sequence. The core idea of PosS is to improve the quality of a single draft path. The paper should discuss PosS's orthogonality with parallel verification methods like tree-drafting.
2. Compared to a single draft model, POSS requires training and storing multiple specialist models. While the paper argues in Appendix C that the inference-time memory overhead is acceptable (approx. 218M per specialist), the training process becomes more complex (e.g., requiring initialization from a pre-trained EAGLE model and subsequent staged training). It would strengthen the paper to briefly discuss this trade-off and its impact on training time and computational resources.
3. The specialist allocation strategy (PosS-n) is straightforward, but its optimality is unexplored. A brief discussion on potential alternative or more adaptive allocation strategies would strengthen the paper.

**Questions:**

1. It accurately identifies a key problem in speculative decoding and proposes an innovative, effective, and well-reasoned solution. The experiments are solid, the analysis is thorough, and the conclusions are well-supported.

---

> ### Author Response · Authors · 2025-11-21
> **Response to Reviewer ATMa**
>
> We sincerely appreciate your recognition of our work. We respond to each comment as follows. We sincerely hope you can reconsider your assessment if you find our response satisfactory.
>
> ---
>
> **W1**: The paper mentions "tree-draft" in Section 6.2 but does not sufficiently clarify its relationship with the experiments. It's unclear whether "draft depth" in Sections 6.2 and 6.3 refers to the depth of the tree or simply the length of a linear draft sequence. The core idea of PosS is to improve the quality of a single draft path. The paper should discuss PosS's orthogonality with parallel verification methods like tree-drafting.
>
> **A1**: We clarify that all experiments exhibited in our paper use **tree-draft** method at inference time, as introduced in *Section 4.1Experiment Setup - Implementations*. Thus, the “draft depth” in Section 6.2 and 6.3 refers to the depth of the draft tree.
>
> You are absolutely right about the orthogonality of linear draft training and tree-draft inference. We have added an explicit statement about the tree-draft inference strategy in Appendix A.2 of our revised paper to avoid misunderstanding.
>
> ---
>
> **W2**: Compared to a single draft model, PosS requires training and storing multiple specialist models. While the paper argues in Appendix C that the inference-time memory overhead is acceptable (approx. 218M per specialist), the training process becomes more complex (e.g., requiring initialization from a pre-trained EAGLE model and subsequent staged training). It would strengthen the paper to briefly discuss this trade-off and its impact on training time and computational resources.
>
> **A2**:
> We clarify that the training complexity of PosS is nearly identical to EAGLE methods.
>
> (1) Initialization from a pre-trained EAGLE model does not mean PosS is trained for more steps. Take EAGLE-3 for example, EAGLE-3 is trained for 10 epochs, and PosS-3(E3) is initialized from the $5^{th}$ epoch checkpoint of EAGLE-3 and continues training for the rest 5 epochs.
>
> Besides, the initialization is as simple as copying parameters from the half-trained EAGLE layer to each of PosS’s layers.
>
> (2) The difference between training code of PosS and EAGLE-3 is minor, which lies only in layer selection, and can be concluded in a few pseudo-codes:
>
> **EAGLE-3**
> ```
> Prediction = eagle_layer(input)
> ```
> **PosS**
> ```
> Selected_layer_idx = position_idx // n # n=3 when using PosS-3
> poss_layer = poss_layers[Selected_layer_idx]
> Prediction = poss_layer(input)
> ```
> Therefore, the extra complexity is rather marginal.
>
> (3) To quantify the impact on training time and computational resources, we provide the training statistics:
> |  | Training Time | GPU Memory Usage |
> |---|:---:|:---:|
> | EAGLE-3 | 1.000x | 63.42GB |
> | PosS-3(E3) | 1.009x | 65.90GB |
>
> ---
>
> **W3**: The specialist allocation strategy (PosS-n) is straightforward, but its optimality is unexplored. A brief discussion on potential alternative or more adaptive allocation strategies would strengthen the paper.
>
> **A3**: The optimal allocation of specialists is one of the extensions to our work. Here are some insights that we can provide for a better understanding of specialist allocation.
>
> (1) The position-specialist layers must preserve their sequential order, because PosS follows a **boosting-style** design: each subsequent layer refines the bias produced by the preceding one. Therefore, if the layers are not used in their relative order, e.g., dynamically selecting layers in an MoE manner, the boosting property is disrupted and decreases prediction accuracy significantly. We design an experiment with standard MoE for demonstration. Here, PosS-3(E3)-MoE applies the same architecture as PosS-3(E3), except for a lightweight router adaptively deciding which layer to use at each position. The results in the table below prove that **the sequential order of layers** is crucial for addressing feature deviation and thus improving prediction accuracy.
>
> |  | Temperature=0 |  | Temperature=1 |  |
> |---|:---:|:---:|:---:|:---:|
> |  | $\tau$ | Speedup | $\tau$ | Speedup |
> | PosS-3(E3) | **5.12** | ***3.27x** | **4.75** | **2.77x** |
> | PosS-3(E3)-MoE | 4.62 | 2.91x | 3.93 | 2.05x |
>
> The complete experiment results are exhibited in Table 8 in our revised paper.
>
> (2) Dynamic number of positions assigned to each specialist. For simplicity, in our paper, we fixed the allocation for each specialist (3 positions for each specialist in PosS-3). However, an adaptive allocation strategy should improve the performance. As implied by Equation (4), earlier layers have a larger impact on the acceptance length. Therefore, the degree of specialization should be allocated more to layers with stronger influence and fewer to those with weaker influence. For example, turning the specialization of 3,3 into 2,4 should be beneficial. A trainable predictor can further improve the allocation strategy.
>
> We have supplemented the discussion on adaptive allocation strategies in **Appendix D** in our revised paper.

---

### Author Response · Authors · 2025-11-21
**Common Response to All Reviewers**

We sincerely thank all reviewers for their insightful feedback. We have revised the paper to address the comments. To summarize our major updates and new experiments:

* **Deployment on vLLM** (response to **dNHR & udoP**): We integrated PosS into vLLM. On Llama-3.1-8B-Instruct, PosS-3 consistently outperforms EAGLE-3. At batch size of 4, PosS-3 achieves a speedup of 2.07x, 26.2% faster than EAGLE-3.
* **Comparison with Gumiho** (response to **dNHR**): We compared PosS with Gumiho, a relevant method with layer specialization. Across different experiment settings, PosS outperforms Gumiho in overall speedup ratio and stability.
* **Dynamic layer allocation, including MoE** (response to **ATMa & udoP**): We explored the way to dynamically allocate layers for PosS. The failure of MoE-based allocation proves the necessity of layer specialization. Takeaway: layers have to be called in sequential order, but the number of positions assigned to each layer is flexible.
* **Ablation on training strategy** (EAGLE-3 + HASS, response to **dNHR**): We experimented with different training strategies. Comparison of PosS-3(E3), EAGLE-3, and EAGLE-3 + HASS demonstrates that PosS mainly benefits from its sequential layer specialization.

---

### Meta-Review · Area_Chair_JS58 · 2026-01-02

**Summary:**

This paper introduces Position Specialists (PosS), a self-drafting speculative decoding approach that instead of a single draft model,  uses multiple, position-specialized draft layers. The main motivation is to improve the drafting accuracy and thus increase the acceptance rates. The authors conduct Extensive experiments on site datasets using Llama-2-13B and Llama-3-8B, and demonstrate noticeable improvements over  baselines methods such as  EAGLE-3.

**Reviewer Concerns:**

The reviewers commented on the novelty of the approach compared to methods like Medusa. Another question was the need for ablation study that disentangled the gains from multi-positional drafters from feature alignment. One of the reviewers also asked about comparison to a single draft  model with an equivalent parameter count (e.g., an MoE). Another review mentioned high-deployment complexity and unfavorable cost-benefit ratio.

**Reviewer Scores:**

udoP 4->6
PRFX 4->6
dNHR 4 ->6
ATMa unchanged;

I believe the authors have mostly addressed  the reviewer concerns. However, the paper still has a serious shortcoming, as it fails to mention an important related approach. In particular, PosS is very similar to Multi-Token Prediction (MTP) suggested in DeepSeek-V3 technical report (https://arxiv.org/abs/2412.19437v1). Specifically, the report suggests using MTP for speculative decoding for accelerating inference. This is a critical omission by the authors, and severely limits the novelty of the contribution.

---

### Decision · Program_Chairs · 2026-01-26

Reject